# Laying the Foundations of Continuing Education in Health in the Family Health Strategy

Cleson Moura [1], Katia Moreira [2], Andreia Costa [3], Cristina Lavareda Baixinho [3], Maria Adriana Henriques [3] and Marcelle Miranda da Silva [1,*]

1    Anna Nery Nursing School, Federal University of Rio de Janeiro, Rio de Janeiro 21941-901, RJ, Brazil; cleson@unir.br
2    Departamento de Enfermagem, Universidade Federal de Rondônia, Porto Velho 76801-058, RO, Brazil; katia@unir.br
3    Nursing School of Lisbon, Nursing Research, Innovation and Development Centre of Lisbon (CIDNUR), 1600-190 Lisboa, Portugal; andreia.costa@esel.pt (A.C.); crbaixinho@esel.pt (C.L.B.); ahenriques@esel.pt (M.A.H.)
*    Correspondence: marcellemsilva@eean.ufrj.br

**Abstract:** Primary healthcare must guarantee health and well-being for the community as a whole, ensuring equity and quality in different responses. For this goal to be achieved, teams must be trained and integrated, and service flows must be functional. Continuing education in health, as a form of training professionals in the workplace, helps to center responses around the needs and preferences of people and families, and to balance the fulfillment of demands by using better work management as a starting point. The objectives of the present qualitative study were to elucidate the meaning given by health professionals who developed their activities in the Family Health Strategy in order to continue education in health, and to discuss the qualification and structuring of work management with this type of education as a background. The methodology used was Straussian grounded theory. Thirty professionals in four Brazilian health units who had experience in the family health field participated in the study. Data were collected between June 2018 and May 2019. Based on three categories, the emerging substantive theory was as follows: laying the foundations of continuing education in health in a collective dialogic and dialectical effort to contribute to the qualification of the work processes in the Family Health Strategy. The professionals' accounts showed that they recognize the importance of continuing education in health and the need to discuss it, given its potential to transform and to assist in the education of professionals with autonomy in the management of their work processes.

**Keywords:** education; public health professional; health promotion; family health; nursing; primary healthcare

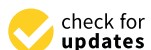

## 1. Introduction

Continuing Education in Health (CEH) is an education process for professional working in health services. It is characterized by educational practices developed in working routines that problematize work processes and provide greater clarity regarding the population's health needs, which results in a balance between demands for and offers of services [1,2].

The theoretical and conceptual bases point to a theoretical–educational proposal for meaningful learning that agrees with Paulo Freire's educational principles, and which features the problematization of the educational environment as a starting point to spark the interest of students. Students must participate in the educational process as active agents in order to make it possible for them to solve problems by means of knowledge construction and deconstruction [1–4].

Therefore, CEH is an important tool used in the management process to qualify work; it is capable of promoting collective spaces for reflection on daily work, guidance for

improving actions, political engagement, and changing reality, as it is only possible to transform what is known [5].

In 2004, CEH was instituted as a state policy in Brazil, to guide the training and qualification of health professionals to the health needs of the population and the development of the national health system. Since then, it has been submitted to successive regulations to strengthen the conditions for its implementation at all levels of healthcare [2,6,7].

Primary healthcare (PHC) stands out among the levels of healthcare as a preferred and ordering gateway with high health problem-solving capacity. The delivery of care depends on the continuous improvement of the work management process and investments, among which is the CEH, linked to the quality improvement cycle [8].

The National Policy of Primary Care recommends the Family Health Strategy (FHS) as a priority model to organize work processes in PHC. Continuing education for an FHS team must be strengthened by managers in the three governmental spheres (municipal, state, and federal), and this education must be a scheduled activity, included in the professionals' working hours, so it is possible to instrumentalize it for programmatic actions [9,10].

However, several issues weaken and threaten the CEH approach, such as a lack of knowledge to plan and start the program, a lack of human resources and structure for the development of activities, and the need for extensive negotiations with the leadership, which involve planning for the sustainability of actions [11].

Education of professionals in the context of PHC has been studied in many countries [12–17], and their realities face common challenges related, mainly, to the absence of conditions and/or culture of the CEH movement, which is reflected in sporadic and circumstantial activities, far from the real needs of professionals and users.

This fact is due to the lack of clarity on the potential of CEH, among which is the discovery of ways to solve problem situations and reorganize health work. These measures are directly related to the quality of care and to the expansion of users' access to health services. However, to achieve these results, it is necessary to have a formal space so that professionals can rethink, articulate, and discuss strategies for the transformation of practices, in the exercise of the action–reflection–action triad [5,15].

Therefore, the following question was formulated: what is the meaning of CEH for FHS teams? The objectives of the present study were to shed light on this meaning and to discuss how CEH can be used to qualify and structure work management in the FHS.

## 2. Materials and Methods

This was a descriptive, exploratory, and qualitative study, guided by the Consolidated Criteria for Reporting Qualitative Research, Straussian grounded theory (GT), and Paulo Freire's educational theories [4,18].

The data were collected between June 2018 and May 2019 in four family health units located in the urban area of Porto Velho, state of Rondônia, Brazil, because of the ease of access which was necessary for collecting and analyzing data simultaneously [18].

Following the concept of theoretical sampling in GT [18], the participants in the first sample group were professionals from FHS teams, guided by the study objective. These participants raised hypotheses about the co-responsibility of CEH, involving professionals from higher levels of management and leadership. To clarify this issue, three other sample groups were developed, always ensuring simultaneity between data collection and analysis. In this progression of the sample in relation to the characteristics of the participants, the initial question was adjusted according to the hypothesis involving each group (Table 1).

The inclusion criteria were as follows: to be a government employee, and to have worked in the same role for at least one year. For professionals from FHS, there was a requirement of being part of complete teams (in accordance with the parameters of the current National Policy of Primary Care) [9]. No participants were excluded as a consequence of being on vacation or leave of any kind. Professionals who were part of the family health unit where the main author worked as a dentist were excluded.

**Table 1.** Sample groups, hypothesis, and initial question in the interview. Porto Velho/RO, Brazil, 2019 (*n* = 30).

| Sample Group | Hypothesis | Initial Question in the Interview |
|---|---|---|
| First (*n* = 14): professionals from FHS teams. | Expected to meet the study objective | In your opinion, what is continuing education in health? |
| Second (*n* = 6): Municipal health managers. | Assigning the responsibility for CEH practices to health professionals and managers. | Comment on the statement: "The responsibility for CEH should belong to health professionals and managers". |
| Third (*n* = 6): leaders of PHC units. | Assigning the responsibility for CEH practices to health professionals, managers and leaders of PHC units. | What should you do for CEH in the context of the FHS? |
| Fourth (*n* = 4): professionals from the CEH Management Center of the Municipal Health Department. | Assigning the responsibility for CEH practices to the CEH Management Center. | Comment on CEH Management Center's responsibility in the process of continuing education of FHS teams, managers and leaders of PHC units. |

This theoretical sampling contributed to data saturation. In addition, two conceptual models related to GT were applied, namely inductive thematic saturation and theoretical saturation [19,20]. In the first model, which refers to the emergence of new codes, it was observed that from the 12th interview no new conceptual codes, categories, or subcategories were identified. In theoretical saturation, which concerns the degree of development of the categories, it was observed that the 22nd interview was the last to contribute to the deepening of the dimensions and properties of an already existing code.

The professionals were invited in person to participate in the interviews. All interviews were conducted by the main author, trained to apply the technique during the interinstitutional PhD in nursing program in Rio de Janeiro. Such programs make it possible to train doctors outside the regions where research is more consolidated in Brazil, based, among other important guidelines, on internationalization. Furthermore, it is noteworthy that in addition to composing an FHS team in Porto Velho, the main author was the coordinator of the Multiprofessional Residency in Family Health, and the occupation of such spaces boosted interest in the theme.

The semi-structured interviews were carried out at the units, outside working hours, in a private and quiet room. The interviews were guided by a question or statement and other scripts that were used to deepen the subject (Figure 1). No participants withdrew from the study and there was no need to repeat any interviews. Non-participant observation was also applied, which originated field notes, memoranda, and diagrams [18].

The data were organized using Microsoft Excel and codified in the following three steps: open coding, axial coding, and integration [16]. The first step consisted of reading the raw data word by word, line by line, or paragraph by paragraph. In axial coding, the preliminary codes were grouped according to their similarity. During the integration step, the theoretical matrix was obtained based on the paradigmatic analysis tool (conditioning factors, action–interaction strategy, and consequences), exposing the subcategories that supported each category and the interrelationship between them and the core category/phenomenon (Figure 1) [18].

The capacity of the theoretical matrix to explain the phenomenon was validated using the following criteria: fit, comprehension, and theoretical generalization [21]. The first validation was carried out with professionals from the CEH Management Center by taking into account the importance of strategic action to develop CEH. One out of the six professionals did not fit the criteria, and another did not answer the invitation sent by e-mail. Only one in the remaining subgroup of professionals sent the material back within 60 days, the established deadline, despite their agreement to participate. The COVID-19 pandemic contributed to this low adherence.

Consequently, the authors opted to expand the process to an external validation. Additionally, because it was a PhD, the participation of professors from the research group was considered convenient. The research group consisted of nine professors, excluding the research supervisor. Three professors were selected because they met the criterion of experience in the subject and/or methodological framework. They received the material in advance. An online meeting was held in November 2020, moderated by the research supervisor. The main author introduced the matrix, followed by an extended discussion.

The engagement with the investigated phenomenon, the dedication of sufficient time to data collection and its continuous review conferred methodological rigor. The principal investigator, responsible for data collection, established a good relationship with the participants. The validation process ensured the accuracy of the findings. In line with grounded theory, the limited literature review at the study design reduced the likelihood of bias in data collection, analysis, and coding. All stages of the study were independently reviewed by professors experienced in the method, theoretical framework, and thematic area, and discussed in a group during the evaluation stages of the doctoral course carried out by the main researcher.

The proposal was approved by a research ethics committee in April 2018, as per protocol no. 2618451. The anonymity of the participants, who signed free and informed consent forms and were aware of their right to withdraw from the study at any time, was guaranteed.

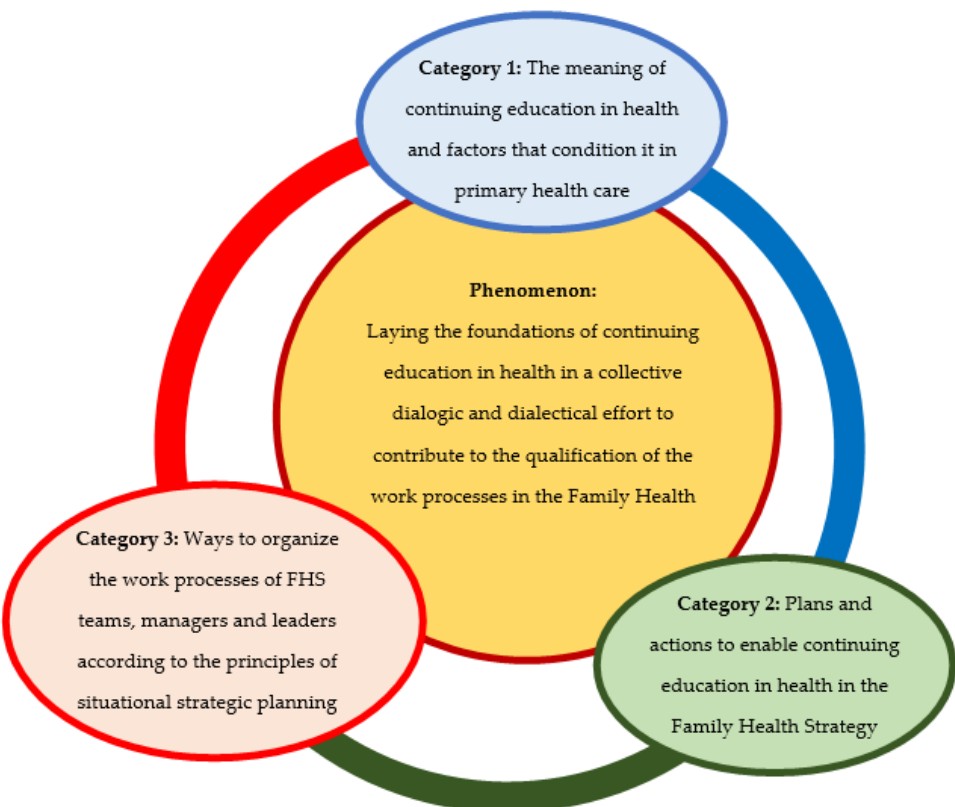

**Figure 1.** Substantive theory and its categories. Source: Designed by the authors.

## 3. Results

Of the 30 participants, 22 were women; 14 were between 31 and 40 years old, 10 were between 41 and 50 years old, 5 were over 51 years old, and 1 was 28 years old. Eighteen had been working in the FHS or the management field for over five years. Five were dentists, three were physicians, seven were nurses, seven had a degree in other health-related areas (nutrition, psychology, biomedicine, and veterinary medicine), five obtained their degree in other areas (administration, social sciences, and advertising and marketing), and three did

not have a higher education degree. Additionally, 22 had graduate degrees, 13 of them in PHC. The four validators were nurses with a PhD, an average age of 40 years, and average length of professional experience of 15 years.

The average duration of the interviews was 25 min. Non-participant observation originated 45 field notes and 16 memoranda and diagrams. The open coding originated 2239 preliminary codes, of which 1747 were used, because they were aligned with the study objectives. The axial coding produced 17 conceptual codes, 10 subcategories, and 3 categories. Grounded in these three categories, the phenomenon was given the following title: laying the foundations of CEH in a collective dialogic and dialectical effort to contribute to the qualification of the work processes in the Family Health Strategy (Figure 1).

Figure 1 demonstrates the relationship between the categories for the development of the phenomenon, which in a few words presents the concept considered the main theme of the study. Each category contributes to the explanation of the phenomenon, with some highlights as follows: category 1 reveals the conceptual problem of the CEH and that it is a process under construction dependent on multiple factors; category 2 values the dialogic process from the collectivity; while category 3 announces, as a consequence of the plans and actions, the improvement in the quality of the work process.

Category 1 was made up of four subcategories (Figure 2).

**Subcategory 1.1:** Recognizing the importance of continuing education in health

**Subcategory 1.4:** Understanding continuing education in health as a dialogic teaching–learning process, despite not being part of work routines

**Category 1:** The meaning of continuing education in health and factors that condition it in primary health care

**Subcategory 1.2:** Relating continuing education in health to sporadic education or to popular education

**Subcategory 1.3:** Pointing out the conditioning factors of continuing education in the context of primary health care

**Figure 2.** Category 1: The meaning of CEH and factors that condition it in PHC.

In subcategory 1.1, CEH was recognized as an important element for professional training, the identification of problems and solutions, and the implementation of knowledge.

> *For me, CEH is very important, because we know that in the area of health, no knowledge is definitive, everything can change very quickly, so CEH becomes a very useful tool for professionals to continue updating in the face of news, of what is emerging for best practices.* (E25)

*[ … ] if professionals improve their knowledge, everyone wins: patient, community, family health unit, manager.* (E2)

Conceptual problems were evidenced, such as those relating the CEH to popular education, or that recognize sporadic education activities as sufficient to characterize CEH, as punctuated in subcategory 1.2.

*In my opinion, continuing education has to do with monitoring patients throughout their lives, working on all aspects, prevention, healing [ … ]. We organize educational talks about everything covering well-being in the health area.* (E20)

*I went to the syphilis course once, which I thought was very good [ … ]. Last year there was a wonderful course on hypertension and diabetes [ … ]. I like when there are activities like courses as CEH.* (E27)

The presence of a motivational leader, availability of resources, and a favorable political and economic context were conditioning factors mentioned in subcategory 1.3. The lack of resources, mainly human, was highlighted, which implies an overload.

*After the nurse began working here. She is stricter, likes to talk, criticize, she tries to make the team work according to a routine, with the members always learning with one another.* (E2)

*It depends on the manager, the professional, but also on the social, political, cultural context [ … ]. Since these days it has been hard at the national level, Porto Velho falls short [ … ]. Ideally, we should have an auditorium, a room, some space in the unit to hold meetings. Although sometimes they take place at the neighborhood association.* [next to the health unit] (E2)

*That this training occurs in the workplace, with reality as a starting point [ … ].* (E7G)

*What makes things difficult is the question of time, we have no time, we are under pressure to provide care to all patients [ … ]. But although it seems that with the CEH we are wasting time, in the long term we will be able to provide much better assistance.* (E30)

The professionals reinforced the evidence that the CEH actions, in addition to happening sporadically and without planning, did not reach all professionals. However, the dialogic perspective appears in the quotes by highlighting the CEH as an opportunity to exchange information, teach, and learn from each other, as observed in subcategory 1.4.

[The] *CEH does not yet include all professionals. It is more common for higher education professionals, dentists, nurses and doctors, who are responsible for multiplying this knowledge with the team.* (E6)

*In my view, CEH is a process that combines the knowledge of the team's professionals, as well as academic knowledge with the empirical knowledge of the population; and we teach people as much as we learn [ … ]; allows us to share knowledge.* (E1)

Category 2 was made up of four subcategories (Figure 3).

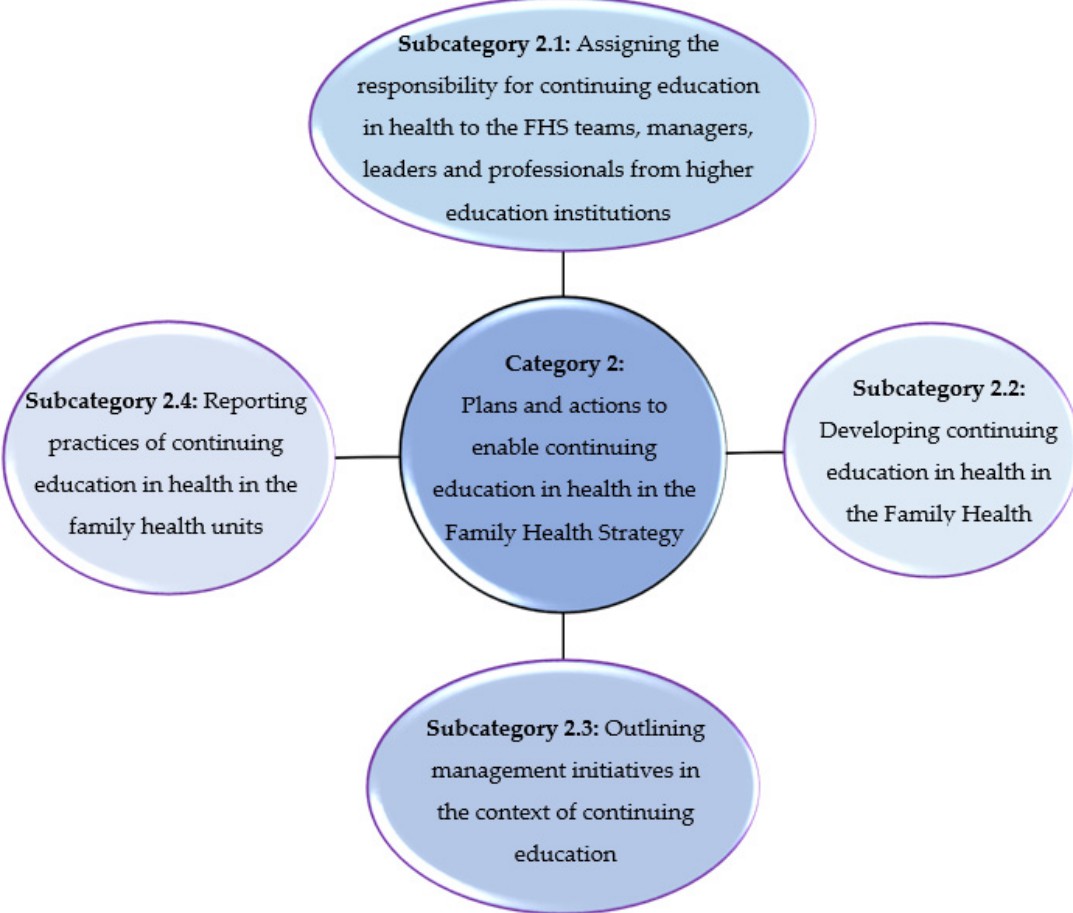

**Figure 3.** Category 2: Plans and actions to enable CEH in the FHS.

Regarding the responsibility of all professionals, it is recognized that CEH is a personal and institutional initiative, and the interest of managers and leaders is indispensable, as provided in subcategory 2.1.

> I believe that if we have interested managers, CEH can be done within the teams [ . . . ]. I think it is also possible for the professional to seek knowledge, because today there are many sources [ . . . ]. (E25)

Regarding the institutional responsibility, it was evident that the CEH actions were under development. Advancements were necessary, such as the institutionalization of a sector to administer these activities. This was one of the most important strategies to develop CEH in the investigated context, grounded in subcategory 2.2.

> We began the activities in this Continuing Education in Health Managing Body two years ago. It was a paradigm shift, with a different kind of work, because before it, education was not continuous. (E15N)

Management initiatives depend on the objective of each division, but the existence of the division that deals with the CEH in each health unit can be favorable to the decentralization of activities, as observed in subcategory 2.3.

> The CEH Management Center has three divisions. One deals with health education in universities and the schedule of internships; the second area has to do with courses and training [ . . . ]. The third division is the NEP, which is the education centers within each health units. The NEP is a way of decentralizing the CEH [ . . . ]. (E16N)

Currently, we have only one NEP working. And we noticed that because it is a scenario that has the university inserted, and that has the proposal of more organized teamwork, it developed CEH activities a little more focused on what we think should be done. (E23N)

In subcategory 2.4, it was highlighted that the creation of the CEH Management Center boosted some practices, such as courses, qualifications, and training, as well as the search partnerships between education and service.

We try to attract educational institutions to become our partners to develop a project that covers the needs of the municipality. (E23N)

There are several ways for CEH to happen [ . . . ]. I've seen some initiatives of small conversation circles; I've participated in training courses [ . . . ]. (E25)

Category 3 was made up of two subcategories (Figure 4).

**Figure 4.** Category 3: Ways to organize the work processes of FHS teams, managers, and leaders according to the principles of situational strategic planning.

The CEH presents characteristics of situational strategic planning in health, such as multidisciplinary teamwork and the problematization of the work context, which allows for a better situational analysis and, consequently, the application of the other FHS guidelines, as observed in subcategory 3.1.

I believe that work in primary care today, in addition to having a definition of territory, of local health diagnosis, and of a group, also needs to have a reason why to do it, for what and for whom the care will be provided. It's organizing, looking at your territory, at your team, and defining where we want to go. Because where you want to go and how to do it are the lines of CEH. With planning it is possible to organize the practice better, and to have a foundation [ . . . ]. CEH provides the answers, guides the development of the work in order to have a result. (E23N)

The CEH makes it possible to organize and guide the work process of the FHS teams, qualifying assistance, improving health indicators and user satisfaction, as punctuated in subcategory 3.2.

The results of education include care with a higher problem-solving capacity, because everyone works in an integrated way. When education is included in the professionals' routine, with the reality of each one as a starting point, the results improve. (E7G)

> [ ... ] when the workshops are over, we can already see results, changes in thinking, in the way of dealing with users, of organizing work, of defining protocols, and this is very interesting. (E5G)

## 4. Discussion

Lack of appropriation of the concept of CEH, which is often mistaken as being either sporadic and circumstantial education, or as being health education, was referred to in the literature [17]. To advance knowledge in the area, it is pertinent to discuss the materialization of the most conservative professional education processes, as illustrated by education that is merely sporadic and circumstantial and loses power regarding the potential for transformation of education and care practices. These processes hinder strengthening the concept of CEH, and even hinder identification of its presence in what can be understood in the context of its informal practices in work routines [6].

The conceptual views of traditional education compromise the institutionalization of CEH. It is necessary to understand the CEH as one that is aligned with the notion of dialogic and problem-raising education, and that it can awaken reflection and critical thinking in students and professionals [4,17,22]. However, managers and care professionals reduce CEH to the offer of and participation in sporadic courses and training sessions, often involving only one type of professional, guided by traditional educational practices to upgrade their knowledge. It is not a matter of depreciating the practices of technical–scientific updating, but of understanding that they should not be the central or even the only purpose of professional education [1,17].

The development of CEH in PHC, a process carried out with continuous feedback, is crucial for implementing health system policy and making it advance, because this educational modality is considered a strategic tool for changing the hegemonic model, which is currently hospital- and physician-centered, and in the process of consolidating the principles of the Brazilian Unified Health System (SUS) [6,17]. The development of CEH in the studied setting meets the process of establishment of SUS in the face of the political and health challenges that pervaded its origins, and institutional trends, some more recent, that are oriented toward professional education in the workplace, especially by engaging health professionals themselves in the institutionalization process.

One of the theoretical constructs of the present study was the recognition of the importance of CEH as a way to train professionals, improve service flows, implement knowledge, and impact quality indicators positively. Although time in the schedule of the professionals for CEH practices has not always been guaranteed, the participants reported that they tried to seek opportunities to be part of educational activities, since they grasped the complexity of nonspecialized care in PHC, which is characterized by the diversity of the knowledge that has to be mastered in order to understand the possible demands that lead users to this gateway to the health system [9,13,15,16,23].

Therefore, educational practices must be part of everyday life in the workplace, preferably by means of active methodologies for meaningful learning [4,17]. It is crucial that the people who engage in these strategies recognize their importance. Some of the actions that stand out in this sphere are conversation circles, team meetings, and periods of integration between education and the service, especially by means of government programs aiming to integrate professors, students, services' health teams, and SUS users. Together, these groups can develop care based on intervention projects guided by situational diagnoses to meet the real needs of the population [17–24].

Teamwork facilitates CEH practices, notably when driven by leaders [22]. Due to the strength of teamwork, evidence indicates that incomplete teams, without representation of a professional category, as recommended by the policy, have difficulties in developing CEH actions and managing the work process. The same happens in teams with high staff turnover, since permanence, congruence, and fluency are required in the implementation of CEH actions [24–27].

However, when the team, even if complete, suffers from human resources that are undersized in relation to work needs, there is work overload, which invariably results in the devaluation of CEH, especially by managers [6,17]. Hence, the influence of the political and economic context, marked by financial crises and cuts in health resources, compromises CEH [10,14]. Lack of investment is a challenge for the institutionalization of CEH and the resulting recognition and professional valorization. Through CEH, plans for positions, careers, and salaries can be implemented to reduce turnover and strengthen professional commitment [16,17].

Dialogue between managers and healthcare professionals in order to discuss the organization of work through continuing education contributes to the care qualification, both in delivering care to the user and in integrated family care, as well in the process management, minimizing, for example, failures in the provision of resources and the user's waiting time for assistance. It should be a concern of managers and leaders to motivate and increase team productivity, by providing opportunities for career construction, and stimulating the feeling of belonging and professional recognition. Motivation and job satisfaction contribute to the stability and development of CEH [24].

Participants recognized the relationship between situational strategic planning and CEH, and the potential of these tools as a basis for the qualification of work processes in PHC and throughout the healthcare network [25–27]. This relationship is corroborated by the recognition of the need to include in the CEH agenda the management and planning tools of the SUS, since one of the weaknesses of CEH is a lack of precise planning [7,17].

One limitation of this study were that it did not include a sample of professionals from educational institutions and family health units in rural areas. Furthermore, only complete teams were included in the study. The turnover of leadership professionals also limited the validation step.

## 5. Conclusions

The professionals recognized the importance of the CEH for the FHS and stressed its relevance in the improvement of work management and, consequently, in improving healthcare quality and disease prevention, but identified problems in its implementation. The CEH process in the investigated family health units is a professional education activity under construction, as it still happens much more in a punctual and contingent way. The professionals asserted that it is necessary to develop the bases for the CEH with interdisciplinary work, recognition of the importance of professional training, the institutionalization of a sector that manages these activities, and strategic planning for better situational analysis and development of management planning instruments.

Everyone must assume responsibilities to initiate and sustain the continuing education process, with the maintenance of dialogue between professionals from FHS teams, managers, and leaders. Professionals recognize that the issue needs to be discussed, given the potential for transforming practices from the exercise of the action–reflection–action triad. In view of the circumstances experienced by the participants, the authors recommend conducting research that applies methods with the potential to accelerate the transfer of knowledge to practice, such as action research, and strategies that favor the sustainability of actions, such as integration between education and the service.

**Author Contributions:** Conceptualization, C.M. and M.M.d.S.; data curation, C.M.; formal analysis, C.M., K.M. and M.M.d.S.; funding acquisition, M.M.d.S.; investigation, C.M.; methodology, C.M. and M.M.d.S.; project administration, M.M.d.S.; supervision, M.M.d.S.; validation, C.M. and M.M.d.S.; visualization, C.M., K.M., A.C., C.L.B., M.A.H. and M.M.d.S.; writing — original draft, C.M. and M.M.d.S.; writing — review and editing, C.M., K.M., A.C., C.L.B., M.A.H. and M.M.d.S. All authors have read and agreed to the published version of the manuscript.

**Funding:** This study was financed in part by the Coordenação de Aperfeiçoamento de Pessoal de Nível Superior—Brasil (CAPES)—Finance Code 001 and by the Center for Research, Innovation, and Development in Nursing, in Portugal, by means of grants provided to some of the authors (CIDNUR_2022).

**Institutional Review Board Statement:** The study was conducted in accordance with the Declaration of Helsinki, and approved by Escola de Enfermagem Anna Nery (protocol code 87525418.1.0000.5238, approved on 24 April 2018).

**Informed Consent Statement:** Informed consent was obtained from all subjects involved in the study.

**Data Availability Statement:** Data are available upon request to the authors.

**Conflicts of Interest:** The authors declare no conflict of interest.

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
