# Peer review of "Laying the Foundations of Continuing Education in Health in the Family Health Strategy"

_education, doi:10.3390/educsci12080521_

Round 1
Reviewer 1 Report
My main concerns:
1. As to the methodology, different groups of participants were asked to comment on different questions based on Table 1. How did that contribute to the development of themes and data saturation in qualitative analysis? Normally the same questions are asked and the common themes emerged from data are reported.
2. In Results section, the authors created some figures to present findings. What's in the figure should be adequately explained in the text with the most relevant quotes. The text and the supporting quotes do not fully match the figures. Figure 1 has 3 categories with one being the substantive category. The text needs to explain this briefly. Figure 2 has 7 subcategories but the explanatory text and quotes do not match all 7 subcategories. Line 167-169 talked about conceptual issues and lack of organization. Which subcategories in the figure are they related to? One of the supporting quotes talked about time, which one is that related to? Same with Figure 3. How do those quotes support what is in the text? Figure 4 has "principles of situational strategic planning" in the title but what are the principles? There are 2 subcategories. Again I don't feel the quotes support the text.
Minor language changes:
1. Line 44: the delivery of care
2. line 62: which is used twice. Reconstruct the sentence to make it clearer.
3. Line 100-102: too many fragments in the sentence. Restructure the sentence.
4. Line 110. Chart 1 - do you mean Figure 1
5. line 136, 138 collection, doctoral. Delete "-"
6. Line 145: 14 and 10 do not add up to 30.
7. Line 151-152: Why the four validators? The inclusion criteria did not mention these 4.
8. Line 159: Diagram 1 - Should it be Figure 1?
9. Figures: Category 1, 2, 3 were not first explained in text.
10.Line 257: stir reflection: maybe "stimulate reflection"?
11. Line 258-260: restructure the sentence.
12. Line 293-295: restructure the sentence
13. line 316: what does "complete teams" mean here?
14. The conclusion can be improved. It should present the issues found in the paper, summarize the main findings and the significance, identify the gap found, the impact of your findings, introduce possible future research on the problem.
Author Response
Response to Reviewer 1 Comments:
We thank the reviewers for their careful reading and for all contributions to the development of this article.
Point 1: As to the methodology, different groups of participants were asked to comment on different questions based on Table 1. How did that contribute to the development of themes and data saturation in qualitative analysis? Normally the same questions are asked and the common themes emerged from data are reported
Response 1:
In the progression of the sample in relation to the characteristics of the participants, the initial question was adjusted according to the hypothesis involving each group. In order to understand the nuances of the context, it was necessary to explore the phenomenon with participants responsible for management and leadership. The confirmation of the hypotheses contributed to reaching the degree of saturation. The questions for the groups that were formed in the process of data collection and analysis in GT allowed the study of hypotheses that emerged from the data.
Point 2: In Results section, the authors created some figures to present findings. What's in the figure should be adequately explained in the text with the most relevant quotes. The text and the supporting quotes do not fully match the figures. Figure 1 has 3 categories with one being the substantive category. The text needs to explain this briefly. Figure 2 has 7 subcategories but the explanatory text and quotes do not match all 7 subcategories. Line 167-169 talked about conceptual issues and lack of organization. Which subcategories in the figure are they related to? One of the supporting quotes talked about time, which one is that related to? Same with Figure 3. How do those quotes support what is in the text? Figure 4 has "principles of situational strategic planning" in the title but what are the principles? There are 2 subcategories. Again, I don't feel the quotes support the text.
Response 2:
We thank the reviewer for the important notice. We have inserted brief texts after each figure to better contextualize them.
“Figure 1 demonstrates the relationship between the categories for the development of the phenomenon, which in a few words presents the concept considered the main theme of the study. Each category contributes to the explanation of the phenomenon, with some highlights: category 1 reveals the conceptual problem of the CEH and that it is a process under construction dependent on multiple factors; category 2 values the dialogic process from the collectivity; while category 3 announces, as a consequence of the plans and actions, the improvement of the quality of the work process”.
In the case of categories, the text has been reorganized to better locate the subcategories.
Point 3: Minor language changes
Response 3: We appreciate the reviewer's attention to correcting spelling problems in the translation. All changes have been made.
About the issue:
Line 151-152: Why the four validators? The inclusion criteria did not mention these 4.
The Straussian Grounded Theory attaches significant value to the theory validation, for scientific rigor and consolidation of study results. Validation is performed to assess whether the theoretical matrix is ​​faithful to the reality studied (adjustment criterion), whether it is understandable and self-explanatory (understanding criterion), and whether it allows for a broad interpretation of concepts (theoretical generalization criterion). There is no specific recommendation on the selection of participants for validation. However, despite the difficulty of carrying out this step with the study participants, we understand the importance of external validation. And because it was a PhD, the participation of professors from the research group was considered constrictive and convenient. The research group consisted of nine professors, excluding the research supervisor, and three were selected because they met the criterion of experience in the subject and/or method. They received the material in advance. An online meeting was held in November 2020, moderated by the research supervisor. The main author introduced the matrix, followed by an extended discussion.
Point 4: The conclusion can be improved. It should present the issues found in the paper, summarize the main findings and the significance, identify the gap found, the impact of your findings, introduce possible future research on the problem.
Response 4:
We chose to transfer the paragraph that was at the end of the results to the conclusion, as its content already met part of what was requested. But we are open to change again, in case this decision was not the best. The final text of the conclusion looked like this:
“The professionals recognized the importance of the CEH for the FHS and stressed its relevance in the improvement of work management and, consequently, in improvement in healthcare quality and disease prevention, but identified problems in its implementation. The CEH process in the investigated family health units is a professional education activity under construction, as it still happens much more in a punctual and contingent way. The professionals asserted that it is necessary to develop the bases for the CEH with interdisciplinary work, recognition of the importance of professional training, the institutionalization of a sector that manages these activities, and strategic planning for better situational analysis and development of management planning instruments.
Everyone must assume responsibilities to initiate and sustain the continuing education process, with the maintenance of dialogue between professionals from FHS teams, managers and leaders. Professionals recognize that the issue needs to be discussed, given the potential for transforming practices from the exercise of the action - reflection - action triad. In view of the moment experienced by the participants, the authors recommend conducting research that applies methods with the potential to accelerate the transfer of knowledge to practice, such as action research, and strategies that favor the sustainability of actions, such as integration between education and service”.
Reviewer 2 Report
Dear Author,
Thank you for your work in this very important topic. You have done a good job using Straussian Grounded Theory in your qualitative methodology design. My only concern, is better integration the quotes into the narrative so the reader can follow how the quotes relate to the category. For example, in the results section, from the end of page 5 through page 8 there is statement, followed by several quotes. If you could comment on how each quote relates, it would be easier to follow.
Author Response
Response to Reviewer 2 Comments:
Point 1: Thank you for your work in this very important topic. You have done a good job using Straussian Grounded Theory in your qualitative methodology design. My only concern, is better integration the quotes into the narrative so the reader can follow how the quotes relate to the category. For example, in the results section, from the end of page 5 through page 8 there is statement, followed by several quotes. If you could comment on how each quote relates, it would be easier to follow.
Response 1:
We are grateful for the careful reading and recognition of the work done. We were very happy with the positive feedback related to the application of GT. In the results, the text has been reorganized to better locate the subcategories and their quotes.

Round 2
Reviewer 1 Report
I appreciate the authors' efforts in revising the manuscript. In the first round of comments, I mentioned that I did not understand the connections between the quotes and themes. I was hoping that I could see clearer connections between those in the revised version. My main concern is still with Results.
Figure 1 gave 3 categories, but which one is Category 1, which one is 2, which one is 3? Can't tell from the figure until we get to the categories.
Category 1 has 7 subcategories which can be simplified greatly. Still how does E5G quote support "importance of CE"? How do E20, E30 quotes support "popular education"? E22G talks about human resources, which subcategory does that support? ....The same issue with Category 2, how does E16N supports the text above "decentralization of activities"?...In Category 3, which quote supports the first subcategory and which one supports the second?
Themes, or categories as you call them, need to be grounded in the data. Each quote should clearly support the theme and the text. I suggest re-develop majors themes and include fewer subthemes. Make sure that the the themes/subthemes/text and quotes match and readers do not have to think hard as to how they support each other.
Author Response
Response to Reviewer 1 Comments (Round 2):
Point 1: I appreciate the authors' efforts in revising the manuscript. In the first round of comments, I mentioned that I did not understand the connections between the quotes and themes. I was hoping that I could see clearer connections between those in the revised version. My main concern is still with Results. Figure 1 gave 3 categories, but which one is Category 1, which one is 2, which one is 3? Can't tell from the figure until we get to the categories. Category 1 has 7 subcategories which can be simplified greatly. Still how does E5G quote support "importance of CE"? How do E20, E30 quotes support "popular education"? E22G talks about human resources, which subcategory does that support? .... The same issue with Category 2, how does E16N supports the text above "decentralization of activities"?... In Category 3, which quote supports the first subcategory and which one supports the second?
Response 1:
Thanks again for the review. We agree with all questions about the organization of the results section. It has been entirely reorganized, as well as the figures have been named subcategories.

Round 3
Reviewer 1 Report
Thank you for providing revisions. I like how you simplified category 1. I like how you organized category 3, pointing back to subcategory 3.1 or 3.2. Maybe you can also do so for category 1 and 2 so that readers can easily figure out the connections.
Author Response
Response to Reviewer 1 Comments:
Point 1:
I like how you organized category 3, pointing back to subcategory 3.1 or 3.2. Maybe you can also do so for category 1 and 2 so that readers can easily figure out the connections.
Response 1:
Thanks again for the review. We fully agree with the suggestion to better reference the subcategories. We organized categories 1 and 2, pointing back to all subcategories.
